# Identifying Factors Predicting Kidney Graft Survival in Chile Using Elastic-Net-Regularized Cox’s Regression

**DOI:** 10.3390/medicina58101348

**Published:** 2022-09-26

**Authors:** Leandro Magga, Simón Maturana, Marcelo Olivares, Martín Valdevenito, Josefa Cabezas, Javier Chapochnick, Fernando González, Alvaro Kompatzki, Hans Müller, Jacqueline Pefaur, Camilo Ulloa, Ricardo Valjalo

**Affiliations:** 1Department of Industrial Engineering, University of Chile, Santiago 8370456, Chile; 2Instituto Sistemas Complejos de Ingeniería, Santiago 8370398, Chile; 3Clínica Santa María, Santiago 7520378, Chile; 4Hospital del Salvador, Santiago 7500922, Chile; 5Hospital Sótero del Río, Santiago 8207257, Chile; 6Hospital Las Higueras, Talcahuano 4270918, Chile; 7Hospital Barros Luco Trudeau, Santiago 8900085, Chile; 8Clínica Alemana de Santiago, Santiago 8320000, Chile

**Keywords:** kidney transplantation, risk prediction, graft survival, regularized models, Elastic Net

## Abstract

*Background and Objectives*: We developed a predictive statistical model to identify donor–recipient characteristics related to kidney graft survival in the Chilean population. Given the large number of potential predictors relative to the sample size, we implemented an automated variable selection mechanism that could be revised in future studies as more national data is collected. *Materials and Methods*: A retrospective multicenter study was conducted to analyze data from 822 adult kidney transplant recipients from adult donors between 1998 and 2018. To the best of our knowledge, this is the largest kidney transplant database to date in Chile. A procedure based on a cross-validated regularized Cox regression using the Elastic Net penalty was applied to objectively identify predictors of death-censored graft failure. Hazard ratios were estimated by adjusting a multivariate Cox regression with the selected predictors. *Results*: Seven variables were associated with the risk of death-censored graft failure; four from the donor: age (HR = 1.02, 95% CI: 1.00–1.03), male sex (HR = 0.64, 95% CI: 0.46–0.90), history of hypertension (HR = 1.49, 95% CI: 0.98–2.28), and history of diabetes (HR = 2.04, 95% CI: 0.97–4.29); two from the recipient: years on dialysis log-transformation (HR = 1.29, 95% CI: 0.99–1.67) and history of previous solid organ transplantation (HR = 2.02, 95% CI: 1.18–3.47); and one from the transplant: number of HLA mismatches (HR = 1.13, 95% CI: 0.99–1.28). Only the latter is considered for patient prioritization in deceased kidney allocation in Chile. *Conclusions*: A risk model for kidney graft failure was developed and trained for the Chilean population, providing objective criteria which can be used to improve efficiency in deceased kidney allocation.

## 1. Introduction

The shortage of deceased donor kidneys available for transplantation has increased the relevance of allocation mechanisms. To better understand the factors predicting transplant outcomes, several survival analysis methodologies have been applied in the literature [1,2,3,4]. The Cox proportional hazards model is the most widely used for assessing the risk of kidney graft failure as it provides comprehensible results that aid clinicians and decision makers. This methodology has been incorporated in the estimated post-transplant survival (EPTS) and kidney donor profile index (KDPI) models used for matching patients with good prognosis after transplant with kidneys with high predicted outcomes in the United States [5,6]. The EPTS considers four recipient factors: age, history of diabetes, history of previous solid organ transplant, and time on dialysis [5]. On the other hand, the KDPI considers ten deceased donor characteristics, including age, height, weight, race, creatinine level, and history of hypertension [6].

In Chile, the deceased kidney allocation system mainly considers HLA mismatch to prioritize patients with good transplant prognosis. Authors have suggested including factors such as donor–recipient age difference [7,8] for longevity matching and for maximizing the utilization of deceased donor kidneys. In this context, it is relevant to provide decision makers with objective information about the effect these factors have shown in historical data from the Chilean population. This may lead to better allocation mechanisms, based on statistical evidence that guides prioritization proportional to the benefit each factor has shown on transplant outcomes. Furthermore, it is of interest to identify which factors are associated with kidney graft survival and to develop a risk model for kidney graft failure capable of revealing improvement opportunities in deceased organ allocation.

Therefore, we have collected data from multiple transplant centers to construct the largest database of kidney-transplanted patients in Chile, comprising 1459 transplants in total. Using this evidence, we have undertaken an in-depth analysis to objectively identify factors influencing transplant outcomes in Chile. However, this sample size is relatively small compared to the number of factors that can be used to predict graft survival [1,6,9]. For this reason, we combined survival analysis with an objective variable selection criterion based on a cross-validated regularized Cox model using the Elastic Net penalty, which provides transparency in the model specification process and could be applied in future studies as more data are collected from transplant centers.

## 2. Materials and Methods

A major data collection effort was conducted in collaboration with five transplant centers in Chile: Hospital Barros Luco Trudeau, Hospital del Salvador, Hospital Las Higueras, Hospital Sótero del Río, and Clínica Santa María, which jointly comprise approximately 14% of the total number of transplants conducted in Chile. To the best of our knowledge, this is the largest kidney transplant dataset that has been collected to date in Chile. This process required the digitalization and standardization of thousands of physical files stored in each medical center. The resulting database was implemented in REDCap and registered 1459 kidney transplant cases. Specifically, it contains pre-transplant information of donor–recipient characteristics and follow-up diagnosis post-transplant, including information on graft failure, with a total of 25 pre-transplant characteristics, which we sought to evaluate as potential predictors of graft survival. Anonymity was preserved for clinical records used in the analysis.

To select the final sample for the study, we applied the criteria summarized in Figure 1. From the 1459 original transplants, 637 entries were discarded sequentially for the following reasons: 244 had transplant dates outside the study period 1998–2018, 12 due to inconsistent recorded survival time or dialysis time, 333 with missing values on variables for which the imputation would be complicated—survival time, HLA mismatch, end-stage renal disease cause, and recipient comorbidities—(later we show how we were able to use these observations to conduct an out-of-sample test), and 48 transplants with pediatric donor or recipient (age < 18). As a result, the final sample comprised 822 transplants, of which: (i) 140 graft failures were observed; (ii) 58 recipients died before graft failure was reported; and (iii) 624 had a functioning graft in the last follow-up.

Table 1 reports summary statistics and the percentage of missing values of the main characteristics collected in the study. Five variables presented missing information that needs to be addressed. To avoid a relevant reduction in sample size and the deletion of relevant information provided by other variables, missing values were replaced by reasonable estimates using MissForest [10]. This nonparametric machine learning algorithm based on random forest has shown good performance imputing clinical data [11].

Two definitions of graft failure are typically used in survival analysis after a kidney transplant [12,13,14]. Overall, graft failure considers the time of transplantation to the date of irreversible graft failure (return to dialysis or a new transplant) or the time of death, i.e., death is considered graft failure. This time may be censored by the date of the last follow-up with a functioning graft. Graft survival censored for death with functioning graft (death-censored graft failure) treats patient death as a censoring event equivalent to the last follow-up, and hence considers a functioning graft at the time of death. The present study used death-censored survival analysis, which more adequately accounts for varying death rates from other causes in the patient population [14].

Preliminary evidence was evaluated through different analyses to better understand the influence between potential predictors and transplant outcomes. First, an exploratory data analysis and a statistical description were computed. Distribution plots for survival and censorship times were examined, and variables were described by their mean and standard deviation or by their frequency and proportion in categories. Second, survival curves were estimated using the nonparametric Kaplan–Meier method for different subsets of patients [15], and the null hypothesis of equality between curves was contrasted using the Mantel–Cox test, considering a 10% significance level [16]. This approximates the survival function, allowing us to assess transplant outcomes in different groups of patients. Finally, penalized splines with 5 degrees of freedom were adjusted in univariate Cox regression for non-binary variables to determine functional transformations that may better fit Cox’s log-linearity assumption [17]. We considered a 10% significance level for detecting statistical evidence of nonlinear effects [18], and we examined complementary graphical evidence for determining adequate functional transformations.

Despite our significant data collecting effort, the sample size was still too limited to evaluate the predictive power of the 15+ factors that needed to be tested, including alternative functional forms. Consequently, we applied an automatized procedure to objectively identify predictors for death-censored graft failure based on a cross-validated regularized Cox model using the Elastic Net penalty [19]. Regularization is a technique for creating generalized models which consists of applying a penalty over the estimated coefficients in order to reduce overfitting and assist variable selection. The Elastic Net penalty is a convex combination of the Lasso and Ridge penalties. Therefore, it combines the strength of the two approaches. On one hand, the Lasso penalty chooses only a few nonzero coefficients, performing variable selection but causing undesirable behavior in the presence of correlated predictors [20]. On the other hand, the Ridge penalty reduces the value of the coefficients towards zero proportionally but sets none to exactly zero; therefore, it better handles correlated predictors but fails to exclude irrelevant variables from the model. The value of α∈[0,1] determines the trade-off between Lasso and Ridge penalties in the Elastic Net. With α = 0.95, the Elastic Net behaves similarly to the Lasso, only removing degenerate behavior due to extreme correlations [21]. We considered this approach (α = 0.95) to prioritize variable selection while maintaining stability in the presence of highly correlated predictors. The optimal penalty in the regularized model was determined by applying 10-fold cross-validation using the Harrell’s Concordance Index (“C-statistic” or “C-index”) as the ensemble metric [22]. To obtain a simplified model, we considered the highest penalty with a C-index no more than one standard deviation from the optimum. This procedure was repeated 1000 times to reduce group selection noise in the 10-fold cross-validation. The selected variables (i.e., those associated with nonzero coefficients) in more than half of the scenarios were defined as the identified predictors by the procedure (see Appendix A). These were used as covariables in a multivariate Cox model to estimate hazard ratios with their respective confidence intervals. Finally, to communicate model predictions beyond hazard ratios, survival curves were estimated for different transplant cases using the Breslow estimator to approximate the cumulative baseline hazard function [23].

Predictive accuracy was assessed by computing the C-statistic, which indicates the proportion of comparable pairs ranked correctly by the model. In addition to in-sample predictions, predictive accuracy was also assessed out-of-sample using 76 kidney transplants not included in the training data due to missing information. These transplants had missing data in recipient comorbidities assessed in the variable selection process. Since some of these were not selected in the final model, this sub-sample was used for evaluating the out-of-sample prediction of the final model specification.

The proportionality assumption in the model was tested using the scaled Schoenfeld residuals [24] and considering the chi-squared distributed statistic proposed by Grambsch and Therneau (1994) [25]. We contrasted the null hypothesis of proportionality considering a 5% significance level, and we examined the graphical evidence of the evolution in time of the scaled residuals for each predictor. On the other hand, the log-linearity assumption was examined in non-binary predictors adjusting penalized splines with five degrees of freedom independently on multivariate Cox regressions [18]. We considered a 5% significance level for null hypothesis rejection and used graphical evidence as a complement.

The statistical analyses were conducted using R (v4.1.2; R Core Team 2021; Vienna, Austria), while data processing was carried out using Python (v3.8.8; Python Software Foundation 2021; Fredericksburg, VA, USA). Detailed information about the libraries and functions used can be found in the Appendix A.

## 3. Results

A visual description of the study cohort is shown in Appendix A. From 822 kidney transplants in the study, 140 (17%) graft failures are observed in a median time of 3.0 (IQR: 0.1–7.4) years from the transplant. An elevated proportion of these occurred in the short term: 48 transplants (34% of total observed graft failures) failed during the first six months after the surgery. On the other hand, 682 (83%) transplants are right-censored with a median censored time of 7.2 (IQR: 4.9–10.2) years from transplant.

MissForest was used to impute missing values in donor history of diabetes (9.8% missing), donor creatinine > 1.5 mg/dL (9.6%), recipient weight (6.8%), recipient years on dialysis (2.5%), and cold ischemia time (0.9%). The statistical comparison between imputed and pre-existing values showed a reasonable estimation by the machine learning algorithm (see Appendix A).

Table 1 shows a statistical summary of donor–recipient characteristics studied as potential predictors for death-censored graft failure. Most kidneys come from deceased donors (80%), male sex (58%), and without history of hypertension (81%). On the other hand, most recipients present history of hypertension (83%), a maximum panel-reactive antibodies (PRA) less than or equal to 10% (64%), and at least two HLA mismatches (88%). In addition, the average recipient is 45 years old, weighs 67 kg, and has spent 3 years on dialysis therapy, while the average donor is approximately 44 years old.

Graft survival at 1, 5, and 10 years from transplant estimated using the Kaplan–Meier method was 93% (95% CI: 91–95%), 89% (95% CI: 87–91%), and 81% (95% CI: 78–85%), respectively. Survival curves were examined in different groups of patients as preliminary evidence and are presented in Appendix A. Considering a significance level of 10%, and without adjusting for other factors, ten variables were identified as potential predictors of kidney graft failure.

The preliminary log-linearity analysis in non-binary variables using penalized splines is shown in Appendix A. A nonlinear effect is observed for recipient age; the risk of graft failure increases at a higher pace beyond 50 years old. The nonlinear component of the penalized spline is significant (*p* = 0.044) at a 10% significance level, while the linear component is not (*p* = 0.732). Considering this evidence, a functional transformation was included to consider an effect for recipients beyond 50 years old. On the other hand, the rest of examined variables showed a reasonable log-linearity adjustment. Nonetheless, three extra functional forms were incorporated considering graphical evidence and medical criteria: a binary variable that becomes active when maximum recipient PRA is greater than 50%, and log-transformations for recipient weight and recipient years on dialysis. These four functional transformations included for variable selection are shown in Appendix A.

The variable selection mechanism is executed, including 25 potential predictors of death-censored graft failure (see Table 2, which includes nonlinear transformations of the variables described in Table 1). Figure 2 shows the number of selected predictors for different penalty values in one iteration of the algorithm. The vertical right line indicates the selected penalty, and the number of selected variables (i.e., with nonzero coefficients in the regularized model) is shown in the top row. The number of times each variable is selected in 1000 iterations is shown in Table 2. Seven variables were selected as predictors by the algorithm: donor age, donor male sex, donor history of hypertension, donor history of diabetes, previous solid organ transplantation in the recipient, the recipient’s years on dialysis log-transformation, and the number of HLA mismatches. The Cox multivariate model adjusted to these factors is presented in Table 3 and presents a C-index of 0.659 in the training data (n = 822) and 0.733 in testing out-of-sample data (n = 76).

The proportionality test shows insufficient evidence to reject the null hypothesis and, therefore, supports the chosen model specification (see Appendix A). Specifically, the *p*-value for the overall test is 0.4, and when evaluating each variable independently, the lowest *p*-value is 0.092 (corresponding to the number of HLA mismatches). In addition, graphical evidence shows a reasonable adjustment of the proportionality assumption (see Appendix A). Correspondingly, the test for log-linearity in non-binary predictors shows a reasonable adjustment (see Appendix A). The nonlinear components were insignificant for donor age, mismatch HLA, and recipient years on dialysis log-transformation.

To further explain estimated effects, survival curves were derived from model estimates using Breslow’s estimator to approximate the cumulative baseline hazard. Figure 3 shows the estimated survival curves for different values in each predictor while keeping the rest constant in the reference values. This graphical evidence provides comprehensible information of model specification under proportionality and log-linearity assumptions.

## 4. Discussion

Regardless of the limitations given by the sample size of the Chilean transplanted population, the results of this study are reasonable considering previous findings in the literature. Donor age and history of hypertension are significant predictors for kidney graft survival identified in populations from the United States, the United Kingdom, and Thailand [6,9,26]. In addition, recipient history of previous solid organ transplant, recipient years on dialysis log-transformation and mismatched HLA have been presented as relevant predictors in several cohort studies [2,3,4] and are included in the EPTS score used for assessing the patient risk of graft failure in the United States [5]. However, regarding the estimated effect for donor sex, which suggests a higher survival in male donor kidneys, this effect could be confounded through other relevant predictors not included in our study due to lack of data—such as donor weight, height, and donor body mass index—which have shown significant associations with transplant outcome in the literature [6,27]. These factors were not included in this study because of missing values in more than 50% of the entries. To analyze whether these omitted variables could be generating a bias in the gender coefficient, we compared means and proportions between donor genders using the Wilcoxon’s test (see Appendix A). Donor weight, height, and cause of death by CVA are statistically different across genders; in particular, male donors have a higher value for weight and height and a higher proportion of deaths caused by CVA. On the other hand, the donor body mass index did not show significant differences between genders. Overall, this analysis suggests that the effect of donor gender could be attributed to omitted variables characterizing the donor; hence, the result should be interpreted with caution. More information about the donor is needed to assess whether donor gender is associated with differences in death-censored graft survival.

Of the six predictors identified, only donor–recipient HLA mismatch is currently considered in the score used to prioritize patients with good transplant prognosis in Chile. In addition, the estimated effects suggest that the current score distinctions between patients may need adjustment to be proportional to the benefits of transplant survival, which may be an opportunity to improve deceased kidney allocation. Furthermore, the five prognostic factors not considered for organ allocation open room for discussion about the mechanisms, based on national evidence, that increase patients’ overall years of therapy.

We compared the predictive power of the model against the C-index that could be attained using the Kidney Donor Profile Index (KDPI) [6]. The KDPI is a numerical measure that combines 10 donor factors to predict graft survival, which is used to match with adequate recipients for kidney allocation. The KDPI is derived from the Kidney Donor Risk Index (KDRI), which is computed using ten factors’ coefficients estimated through a multivariable Cox proportional hazards regression using a sample of deceased donors in the United States. These factors include donor’s age, height, weight, ethnicity, history of hypertension, history of diabetes, cause of death, serum creatinine, hepatitis C virus (HCV) status, and donation after circulatory death (DCD) status.

We applied the KDRI score with the patient sample analyzed in this study, calculating the C-index to assess its predictive power. Since height, weight, HCV status, and DCD exhibited significant missing values in the data, these factors were set at reference levels to compute the KDRI. The predictive power of this adjusted KDRI score yields a concordance index (C-index) of 0.606, which is lower than the C-index calculated in this study (C-index = 0.659). This comparison provided further support that the methodology developed in this work is useful to improve predictions of graft survival, which can potentially be used to construct a “local” KDPI score in Chile’s kidney allocation system.

## 5. Conclusions

This study analyzed the largest sample of kidney transplants collected up to date in Chile, comprising 1459 transplants from five transplant centers accounting for 14% of the transplants performed from 1998 to 2018. An automated procedure based on the Elastic-Net-regularized Cox’s regression was applied to objectively select predictors for kidney death-censored graft failure in the Chilean population, and a risk model that provides comprehensible results was developed to aid clinicians and decision markers. It is relevant that the data collection process continues to increase the accuracy of the selected variables and the precision of the estimated effects through the proposed methodology. In this regard, the procedure can be used as more information is collected if proportionality and log-linearity conditions for the Cox model are adequately validated.

## Figures and Tables

**Figure 1 medicina-58-01348-f001:**
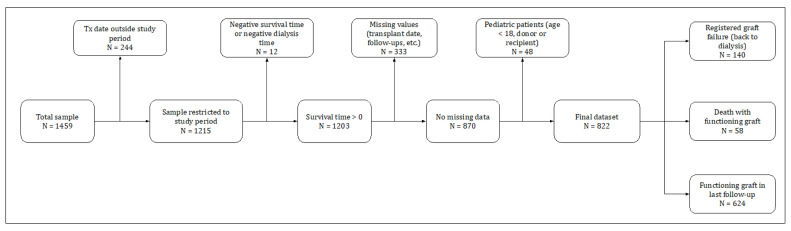
Description of the sample selection process. Selection criteria included cleaning entries with imprecise data, missing values, and focusing on adult patients only.

**Figure 2 medicina-58-01348-f002:**
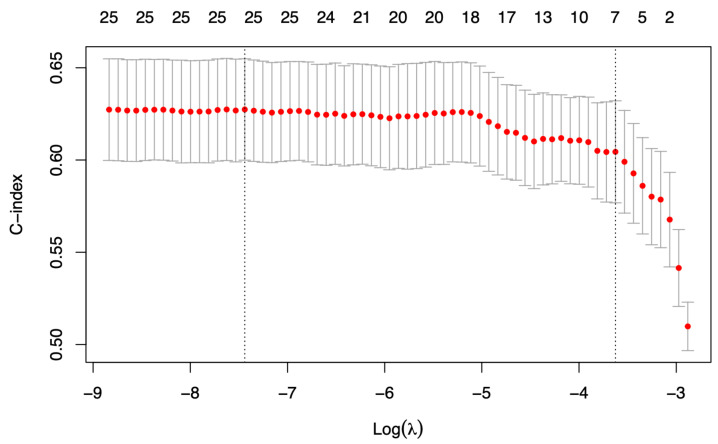
Cross validation plot for variable selection in the Elastic-Net Cox model. The top row represents the number of nonzero coefficients per penalty value. The red dots describe the C-Index associated to each penalty value. The vertical left line indicates the optimal penalty, which maximizes the ensemble C-Index. The vertical right line corresponds to the largest penalty value related to a C-index value within one standard deviation of the maximum C-Index.

**Figure 3 medicina-58-01348-f003:**
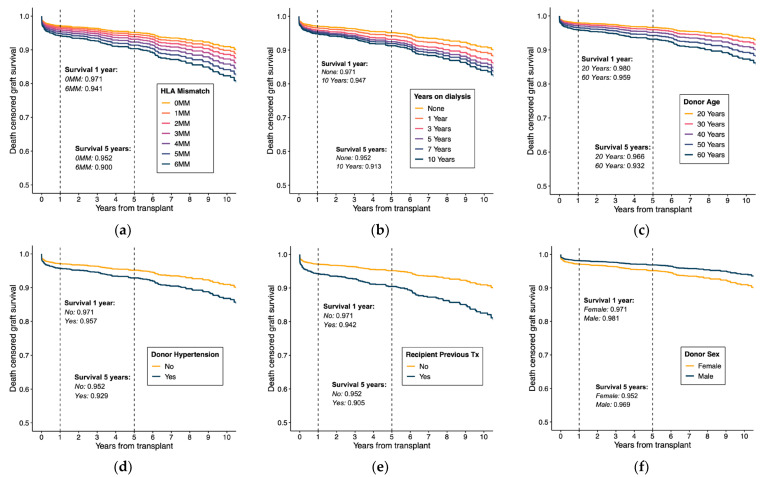
Estimated survival curves derived from the model using the Breslow estimator to approximate the cumulative baseline hazard. Each subfigure shows the survival curves when varying values in one predictor while maintaining the rest of the variables in their reference values (see Table 3). The subfigures correspond to: (**a**) HLA mismatch; (**b**) Years on dialysis; (**c**) Donor age; (**d**) Donor history of hypertension; (**e**) Recipient previous transplant; and (**f**) Donor sex.

**Table 1 medicina-58-01348-t001:** Statistical description of the most relevant potential predictors for death-censored graft failure and their percentage of missing values (which were imputed to conduct the statistical analysis). The study sample included 822 patients. CVA: cerebrovascular accident; HLA: human leukocyte antigens; PRA: panel-reactive antibodies.

Origin	Variable	Mean (SD)/n (%)	Events	Missing (%)
	Age	44.62 (12.91)	–	0.0%
	Weight in kilograms	66.78 (12.62)	–	6.8%
	Years on dialysis	3.16 (2.78)	–	2.5%
	Male Sex			0.0%
	Yes	463 (56%)	78	–
	No	359 (44%)	62	–
	Previous Transplant			0.0%
Recipient	Yes	55 (7%)	15	–
	No	767 (93%)	125	–
	Hypertension			0.0%
	Yes	682 (83%)	111	–
	No	140 (17%)	29	–
	Max PRA			0.0%
	0–10%	529 (64%)	93	–
	11–50%	117 (22%)	25	–
	51–100%	116 (14%)	22	–
	Age	43.88 (13.03)	–	0.0%
	Male Sex			0.0%
	Yes	475 (58%)	66	–
	No	347 (42%)	74	–
	Living donor			0.0%
	Yes	164 (20%)	21	–
	No	658 (80%)	119	–
	Hypertension			0.0%
Donor	Yes	158 (19%)	38	–
	No	664 (81%)	102	–
	Diabetes			9.8%
	Yes	26 (3%)	8	–
	No	796 (97%)	132	–
	Creatinine > 1.5 mg/dL			9.6%
	Yes	36 (4%)	7	–
	No	786 (96%)	133	–
	Cause of Death: CVA			0.0%
	Yes	346 (42%)	68	–
	No	476 (58%)	72	–
	Cold ischemia time in hours	15.62 (9.50)	–	0.9%
	Mismatch HLA			–
Transplant	4–6 MM	282 (34%)	54	–
	2–3 MM	445 (54%)	77	–
	0–1 MM	95 (12%)	9	–

**Table 2 medicina-58-01348-t002:** Number of times each variable is selected as a predictor in N = 1000 iterations of 10-fold cross-validation of the Elastic Net Cox model. ESRD: end-stage renal disease; CVA: cerebrovascular accident; PRA: panel-reactive antibodies; HLA: human leukocyte antigens.

Variable	Type	Selected Count	Predictor
Donor Hypertension	Binary	1000	Yes
Donor Age	Integer	995	Yes
Donor Male Sex	Binary	919	Yes
Recipient Previous Transplant	Binary	822	Yes
Number of mismatch HLA	Integer	822	Yes
Donor Diabetes	Binary	796	Yes
Recipient ln(Years on dialysis + 1)	Numeric	743	Yes
Donor Death Cause: CVA	Binary	385	No
Cold Ischemia Time	Numeric	361	No
Recipient Hypertension	Binary	196	No
Recipient Max PRA > 50%	Binary	142	No
Recipient ESRD Cause: Glomerulopathies	Binary	136	No
Recipient Age	Integer	136	No
Donor Creatinine > 1.5 mg/dL	Binary	126	No
Recipient ESRD Cause: Diabetes	Binary	121	No
Recipient Max {Age—50, 0}	Integer	121	No
Recipient Vascular Peripheral Disease	Binary	112	No
Recipient ESRD Cause: Hypertension	Binary	94	No
Recipient Weight	Numeric	76	No
Recipient Diabetes	Binary	2	No
Recipient Male Sex	Binary	2	No
Donor Living	Binary	0	No
Recipient Max PRA	Integer	0	No
Recipient Years on dialysis	Numeric	0	No
Recipient ln(Weight + 1)	Numeric	0	No

**Table 3 medicina-58-01348-t003:** Multivariate Cox model for death-censored graft failure.

**Variable**	**HR** ^1^	**95% CI** ^2^	***p*-Value**
Donor Male Sex (ref = Female)	0.64	0.46, 0.90	0.010
Recipient Previous Tx (ref = No)	2.02	1.18, 3.47	0.011
Donor Age (ref = 40)	1.02	1.00, 1.03	0.020
Recipient ln(Years on Dialysis + 1) (ref = 0)	1.29	0.99, 1.67	0.055
Donor Diabetes (ref = No)	2.04	0.97, 4.29	0.059
Donor Hypertension (ref = No)	1.49	0.98, 2.28	0.065
Mismatch HLA (ref = 0 MM)	1.13	0.99, 1.28	0.068

^1^ HR = hazard ratio, ^2^ CI = confidence interval.

## Data Availability

Data are available from the corresponding author upon request.

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
