# Peer review of "Identifying Factors Predicting Kidney Graft Survival in Chile Using Elastic-Net-Regularized Cox’s Regression"

_medicina, 2022, doi:10.3390/medicina58101348_

Round 1

Reviewer 1 Report

 Thanks for giving me the opportunity to review this research paper.

My important observations are:

1. Researchers have to understand difference between ' death censored graft failure" and " graft loss or graft failure". They have used these terms as one at different places. Death censored graft loss is if a recipient dies with a functioning graft OR a failed graft where as graft failure alone is different. Isolated graft failure can happen in a living patient and there are different factors for both of them as per previous data. ( Although  a lot of the covariates overlap). We cannot interchange these terms.

Please see the following studies for reference:

Taber DJ, Gebregziabher M, Payne EH, Srinivas T, Baliga PK, Egede LE. Overall Graft Loss Versus Death-Censored Graft Loss: Unmasking the Magnitude of Racial Disparities in Outcomes Among US Kidney Transplant Recipients. Transplantation. 2017 Feb;101(2):402-410. doi: 10.1097/TP.0000000000001119. PMID: 26901080; PMCID: PMC4991956.

Mayrdorfer, M., et al. (2021). "Exploring the Complexity of Death-Censored Kidney Allograft Failure." Journal of the American Society of Nephrology 32(6): 1513.

If we are talking about the death censored graft loss (DCGF/DCGL) then we have to account for many factors that are a common cause of DCGF/DCGL based on the previous research to create a model.

 If we want to create a model you have to consider:

Recipient CVS risk factors : history of CAD , smoking ( I see u included PVD and DM ), 

Recipient malignancy history

Recipients race

Donor : most important is DCD ( donation after cardiac death) , donors history of diabetes

Txp : DGF ( delayed graft function) and many post transplant features ( which I think if u just aim to make an early model might be OK to skip)

Other observations; What is ERC in table 2? All tables  should have abbreviation explanation at the bottom 

The researchers dont have a separate discussion part. If its not required for journal thats fine but ususally its a part of scientific wring where they should elaborate their strengths and limitations in details.

Sounds like a lot of effort placed on models statistically which is great but scientifically we have to keep most important causes of an outcome into account. 

Author Response

We thank the reviewers for the positive feedback and for the important suggestions to improve our work. We have worked hard to address all the comments presented in the revision. The main changes include:

  • A more accurate definition of the methods used for the estimation, the data collection process, and the sample selection criteria.
  • Incorporated additional covariates in the model based on the suggestions provided. This required imputation due to missing values.
  • Additional analysis to evaluate the performance accuracy, including out-of-sample predictions and benchmarking with KDPI methods.
  • Improvements in the writing, organization, and tables/figures.

We hope that this improvement addresses the issues that were presented. Below we include a detailed response to each of the comments suggested by the reviewers. Thank you for your help in improving the manuscript, which we think can have significant potential in Chile to inform the design of the national kidney allocation system.

Reviewer 2 Report

Overall, the paper is well constructed and the methods used are sound. My comments are mainly based on possible ways to improve the impact of the paper:

-       As explained in the introduction, there are already a few algorithms out there to attempt at anticipating outcome, such as KDPI and EPTS. The authors would improve the relevance of their work by comparing their model to these two, in order for the reader to gauge the impact of such research

-       The Cox regression is indeed well suited to anticipate a continuously changing risk, but unfortunately this closes the door on other models, such as classification (plenty of machine learning possibilities here). Have the authors attempted to create binary outcome (e.g. : graft loss at 1 yr, or 5 yr, etc…) and then train classification models ? I would be curious if good accuracies could be derived from these

-       The decision to not perform a train/test split is for me a mistake. Overfitting is almost a guarantee in machine learning, even with cross validation. Testing the relevance of a model on data it’s never seen is in my opinion a necessity. 800+ patients should be plenty to perform such split

-       The authors mention that the influence of donor sex could be confounded by IMC, why did they not check for it? Surely IMC is known? Moreover, is sex matching between donor and recipient considered?

-       Your table does not cite cold ischemia time. Have you not considered this parameter ?

Minor remarks:

-       Correlations tests were Pearson’s, however this test is for normally distributed data. Was that the case for all parameters? Perhaps consider a non-parametric option.

-       Figure S7 is in Spanish. Other figures are also in spanish

Author Response

(The authors gave the same response as above.)

Reviewer 3 Report

Dear authors, thank you very much for your work. It would be very interesting to examine whether collinearity exists within your data and present the results in the methods/suppl. sections accordingly.

All of my best regards.

Author Response

(The authors gave the same response as above.)

Round 2

Reviewer 2 Report

The authors have adressed my remarks to my satisfaction

Author Response

We thank the reviewer for the positive feedback and help in improving the manuscript, which we think can have significant potential in Chile to inform the design of the national kidney allocation system.